# African Swine Fever Status in Europe

**DOI:** 10.3390/v11040310

**Published:** 2019-03-30

**Authors:** Przemyslaw Cwynar, Jane Stojkov, Klaudia Wlazlak

**Affiliations:** 1Department of Environmental Hygiene and Animal Welfare, Wrocław University of Environmental and Life Sciences, Chelmonskiego 38C, 51-630 Wroclaw, Poland; klaudia.wlazlak@upwr.edu.pl; 2Animal Welfare Program, Faculty of Land and Food Systems, University of British Columbia, 2357 Mall, BC V6T 1Z4, Canada; j.stojkov@alumni.ubc.ca

**Keywords:** African Swine Fever, wild boar, domestic pig, Poland

## Abstract

African Swine Fever (ASF) is a highly contagious disease that affects the domestic pig and wild boar population. The aim of this study was to describe the introduction and spread of the ASF virus in Western Europe (1960–1995) and in Eastern Europe (2007–2018), with particular emphasis on the current ASF situation in Poland and its challenges and future perspectives. The first ASF outbreak in Europe was reported in Portugal in 1957, with the virus spreading over most of Western Europe over the next 30 years. In Eastern Europe, the virus was first observed in Georgia in 2007, from where the disease spread quickly to other neighboring countries, reaching Poland in 2014. Since then, there have been 3341 confirmed cases in the wild boar population in Poland. Although there have been no confirmed cases of wild boars coming into contact with domestic pigs, the first notified case concerning domestic pigs was reported in July 2014. Since then, there have been a total of 213 confirmed outbreaks of ASF on Polish pig farms. Given the virulence of the ASF virus and the myriad of transmission routes across Europe, the monitoring of this disease must be a priority for Europe.

## 1. Introduction

African swine fever (ASF) is a highly virulent arbovirus belonging to the Asfarviridae family, genus Asfivirus [1,2]. The disease appears to manifest itself differently in domestic and feral pigs, depending on the virulence of the virus, the infectious dose, and the infection route [3]. Peracute and acute clinical forms of ASF are accompanied by high morbidity and mortality that can reach 100% [3]. Mortality is reduced when ASF is considered to be subclinical or chronic: ranging from 0–60% [3,4]. The shedding of ASF in clinically recovered animals appears to last, on average, up to 6 weeks [3], but there have been reports of longer periods [4], which likely contributes to the persistence of the disease [3]. Early diagnosis of the first cases within a region is imperative, as this aids in the rapid implementation of control measures necessary to control and eradicate the disease [5].

The main route of infection in domestic and feral pigs is via oral or nasal contact with other infected pigs or contaminated materials [6,7]. Given the introduction of free ranging pig systems, ASFV transmission from wild boars to domestic pigs has been reported [6,8,9] to have increased, given the lack of emphasis placed on biosecurity in these types of systems [3,7,8,9]. Other vectors of transmission are the transport of infected animals or products (i.e., pork meat, fodder, wastes) contaminated with the virus [8,10], as well as insects (Ornithodoroserraticus) [4,5,11]. Work on ASF detection has clearly demonstrated that passive surveillance is more effective than active surveillance in detecting ASFV-infected wild boars or domestic pigs [9].

Since there is no dedicated veterinary treatment and no vaccine available for ASF, rapid and specific diagnostic procedures are essential components of the control and eradication of the disease [12,13,14]. Given the similarity between ASF and other swine infections, such as classical swine fever (CSF), porcine dermatitis and nephropathy syndrome (PDNS), porcine reproductive and respiratory syndrome (PRRS), salmonellosis, and erysipelas [4,5,10], there is a need to confirm suspected cases in wild boars and domestic pigs using specific laboratory tests. The World Organisation for Animal Health (OIE) recommends ASF virological diagnosis through a combination of tests including detection of viral genome by PCR (widely used in national reference laboratories), detection of viral antigens by antigen ELISA, or a fluorescent antibody test (FAT), and virus detection using virus isolation [5,12].

ASF was first diagnosed in Kenya in 1909, and reported by Montgomery (1921) as a disease distinct from CSF [3,14,15,16]. The various transmission routes of ASF, such as through direct and indirect contact with infected pigs, contaminated materials and food, and Ornithodoros ticks, contributed to a fast spread of ASFV across the African continent and the subsequent introduction of ASFV in Europe. Since the 1960s, ASF outbreaks have been reported in Europe in two separate incidents. The first spread (1960–1995) was through Spain and Portugal to other countries in Western Europe, and resulted in the eradication of ASF. The second introduction of ASFV (2007–ongoing) was through Georgia, which was considered to be a fundamental viral base, and later entered the Eastern European countries.

The present epidemiological situation of ASF is of great concern for EU countries as it differs from past epidemics. Previously, the transboundary spread of ASFV had been attributed to single viral introductions, facilitated primarily by movements of domestic animals and infected products [17]. Current measures implemented to control ASFV are based on rapid and accurate disease diagnosis, immediate slaughtering, and safe disposal of infected animals [3,4,17,18]. The efficacy of how ASF outbreaks in Eastern European countries such as Poland are managed may have significant epidemiological implications for neighboring countries, which in turn may have profound effects on European pig production and related economic consequences [2,8,19,20].

The first objective of this study was to present the ASF epidemiological situation in Europe, dividing the outbreaks into two distinct chronological periods: the Western European period that began in 1957 and ended in 1995, and the subsequent Eastern European outbreak that occurred between 2007 and 2018. The second objective was to describe the current epidemiological situation of ASF in Poland, following the first documented case of ASF reported in February 2014 to the present day (December, 2018).

## 2. Epidemiological Situation of ASFV in Europe

### 2.1. First Spread of ASFV in Europe (1960–1995)

ASFV was first observed in Portugal in 1957 and then it spread quickly to the surrounding Western European countries, with Spain being the first to report cases, followed by Italy, France, Malta, Belgium, and the Netherlands [3,11,17]. However, by utilizing rigorous disease control programs (e.g., eradication, improved biosecurity measures, education), countries in Europe successfully eradicated the disease in 1995. The exception was Sardinia, which continued to struggle with this disease [8].

#### 2.1.1. Portugal and Spain

The movement of ASFV to Portugal is thought to have taken place via contaminated food waste coming from African airline flights and/or ships docking at seaports, with which pigs were later fed [16,21]. Other vectors that may have contributed to the transmission of ASFV include soft ticks (Ornithodoroserraticus). The virus remained endemic in the Iberian Peninsula until 1995, affecting both domestic pigs and Eurasian wild boars [21,22].

From 1978 to 1995, an increased incidence of the disease was documented in Portugal and Spain [16,23]. Nevertheless, the probability of the virus persisting in the environment is still high because of inadequate biosafety conditions in backyard farms, the presence of large populations of Ornithodoros sp. ticks as long-term ASFV reservoirs, and the uncontrolled wild boar population in Doñana National Park, in the Huelva region, one of the largest national parks in Spain [7,22]. It should also be noted that only 5.8% of ASF outbreaks in domestic pigs in Spain until 1981 were associated with potential contact between domestic pig populations and infected wild boars, as the population of these wild animals is lower than in central or eastern European countries [22]. Since 1994 and 1999, there have been no new confirmed ASF outbreaks in Spain and Portugal, respectively [7,22,24], as the eradication of affected herds and wild boars hunts were established simultaneously with higher biosecurity levels on pig holdings. ASF was finally eradicated from the Spanish domestic pig population in 1995, after 35 years of significant effort [22].

#### 2.1.2. France, Italy and Malta

Given the confirmed presence of ASF in Portugal and Spain, it was not surprising that cases were identified in France and Italy, but this did not affect the animal production significantly. There were three confirmed ASF outbreaks in France (1964, 1967, 1974) and in Italy (1967, 1969, 1993), followed by a successful eradication process by wild boar hunting and euthanasia of affected pig herds. The exception was the endemic territory of Sardinia [11,16,19,25]. In Malta, eradication following the 1978 outbreak was achieved by slaughtering the entire swine population. The Pyrenees Mountains were viewed as a natural barrier minimizing the spread of ASFV by known vectors (i.e., wild boars and soft ticks) between Spain and France. Similarly, the Alps between France and Italy were also thought to reduce the number of ASF cases in France [11,16].

#### 2.1.3. Sardinia

The notable exception where eradication did not take place is the Mediterranean island of Sardinia (Italy). This is a location in Western Europe with an endemic form of ASF since it was first identified in 1978, affecting both domestic and feral pigs [8,11,19,25,26]. The fact that more than 70% of the pig population is kept in extensive systems and backyard farms, combined with the close proximity of wild boars may explain why the disease persists. In addition, the European wild boar has free access to grazing pastures and the domestic pig environment. Combined, these are the two main factors that contribute to the ineffective eradication programs on the island [27,28]. Despite the persistence of ASF in Sardinia, there is no evidence linking its presence with pig health and production elsewhere in Europe. Moreover, there is no correlation between the present East European spread of the disease and ASFV in Sardinia [8,11,19,25,26].

#### 2.1.4. Belgium

The first case of the ASF virus in Belgium was reported in West Flanders in March 1985. Although its origin was not confirmed, pork meat had been imported directly from Spain and thus this was thought to be the most probable source of infection. A well organized prevention program, consisting of the elimination of affected pig herds, was immediately implemented. In total, 185 pig holdings were tested, with 12 farms initially testing positive. This outbreak resulted in the slaughter of over 34,000 animals housed on 60 holdings. After serological analysis of 3008 pig holdings (116,308 blood samples), an official report on the ASFV eradication (killing affected animals) was published in September 1985 [11,29].

#### 2.1.5. The Netherlands

The Netherlands was also one of those Western European countries to report an ASF outbreak in the period beginning in 1960 and ending in 1995. Initially, ASF was detected in 1986 near The Hague, in the South Holland region, with transmission being attributed to food wastes arising from public buildings (hotels, restaurants, hospitals) that were used in animal feed. Although the presence of ASFV was confirmed within 3 weeks after the first clinical signs were observed, the outbreak resulted in the depletion of 19% of the pig population in the region [11,30].

### 2.2. Second Spread of ASFV (2007–2018)

The first known case of ASF was recognized in Georgia in 2007, which was followed by numerous outbreaks of ASF among domestic pigs and wild boars in the Russian Federation (2008), Ukraine (2012) and Belarus (2013) [3,7,10,25]. This rapid infiltration of ASFV in East European and West Asian countries most possibly occurred due to the lack of emphasis placed on ASF contingency plans and preventive measures [11].

#### 2.2.1. Georgia

Following the initial diagnosis of ASF on 7 June 2007, several outbreaks in Georgia have been confirmed. Although initially the disease in pigs was attributed to porcine circovirus, tests carried out by the OIE Reference Laboratory in Pirbright, United Kingdom, confirmed the presence of ASF. The 11 officially confirmed new outbreaks of ASF in domestic pigs indicated the first occurrence of the disease in this part of Europe [24]. Preventive action and disease control measures, including culling of infected and in-contact animals, and isolation of suspected properties, were immediately implemented by the Georgian authorities. Despite these efforts, over 19% of the domestic pig population (~60,000 pigs) was rapidly infected within the first two months of the presence of ASFV and the animals were euthanisted [19].

The vehicle of transmission is thought to be the use of contaminated meat products (later fed to pigs) from international ships arriving at the Black Sea Port of Poti in the Samegrelo-Zemo Svaneti region [10,19,30]. The specific origin of the Georgian ASFV is still unknown, although some findings link the virus genotype with that in Madagascar and Mozambique [12,25,31]. However, despite efforts to minimize the spread of the disease and ultimately eradicate it, ASFV still persists in Georgia. The primary reason ascribed to this failure is the low on-farm biosecurity [7,11,24,25]. The disease spread from Georgia and affected Abkhazia and South Ossetia, and was also recently reported in Armenia and Azerbaijan [7,18,32]. ASF virus was detected in dead wild boars in the Russian Federation in September 2007 and spread widely in the Caucasian region, where it is now viewed as an epidemic [7,18,32].

#### 2.2.2. The Russian Federation

Following the initial outbreak in the 1970s, which was quickly eradicated, the first ASFV case in the last decade was identified in November 2007 and reported to the OIE on 4 December 2007. The primary vehicle of transmission is thought to be the presence of 15 dead wild boars that were registered in Shatol’skoe Ushchel’e along the Argoun River in Shatoysky region, Chechnya, an area that shares a border with Georgia [19,33]. The presence of this intensive and permanent ASFV transmission among the wild boar population in infected districts of the Chechen Republic is thought to be the primary reason for the virus spreading to the North Caucasian region, where it is now circulating among the wild boar population in the territories of Ingushetia (June 2008), North Ossetia (June 2008), Kabardino-Balkaria (December 2008), and Dagestan (September–October 2009 and March 2010) [10,26]. The first outbreaks in domestic swine were reported in June 2008 in North Ossetia [26,34]. In January 2009, ASFV was detected in wild boars in the Stavropol region, over 150 km from the first outbreaks within Russian Federation. In the same region, a total of 177 outbreaks were reported over three consecutive years [26]. The presence of a wild boar population, and the low adherence to on-farm biosecurity in the North Caucasus region may explain the tremendous spread of this disease. Across all districts of the Russian Federation, there have been a total of 52 recorded ASF outbreaks up to February 2009.

Not surprisingly, given the close proximity of the wild boar population, traditional backyard farms were affected the most; with 34% of RF pig production affected, this is where the majority (63%; 41.855 pigs) of ASF outbreaks occurred [7,10,18,26]. In addition, the free-range farming systems increased ASFV transmission between the wild boar population (94% hosts of EU ASF) and domestic pigs. This produced a positive feedback effect on the virus circulation, which created tremendous challenges that prevented eradication [17]. Since 2011, the disease has moved north, reaching St. Petersburg, Archangel and Murmansk, which had previously been ASF free. Since then, over 20 notified outbreaks have been reported [26]. These outbreaks have affected both wild boar and domestic pig populations, creating new endemic areas, such as the Tver region, located 150 km from Moscow [17]. Since 2012, the disease has spread across the Krasnodarskiy region [26], located in the Russian Southern Federal District, above the Black Sea, indicating intensive dispersion of ASFV. In 2013, numerous outbreaks were registered in the western part of RF, including Smolensk and Pskov regions bordering Belarus, which was a significant issue for EU member states.

By 14 September 2016, RF had reported 110 ASF outbreaks to OIE [32]. The European Food Safety Authority (EFSA) has reported that, since 2007, about 40% of all registered ASF outbreaks in RF have been in wildlife. A characteristic feature of the ASFV circulation in RF observed since 2012 are the numerous so-called “infected objects” (i.e., wild boar infected animals) in the environment. In addition, 93 out of the 106 ASF outbreaks in wild boar in 2013 were registered as “infected objects”, but, as they were not considered by the official veterinary authorities as outbreaks, these “infected objects” were not reported to the OIE [18]. Moreover, the insufficient control and eradication measures undertaken by the RF authorities against ASF indicate a lack of interest in disease eradication [10].

#### 2.2.3. Ukraine

The first official ASF outbreak in Ukraine notified to OIE occurred on 30 July 2012 in the Zaporozhye region, near the Black Sea [3,7,17,25]. The death of three domestic pigs on a backyard farm was reported and immediate government action was initiated by euthanised animals in affected areas and establishing quarantine zones [7,19,33]. EFSA reports indicate that the second ASF case was confirmed in a dead wild boar found by the Derkul riverside in Rostovskaya region on 5 January 2014 close to the border with RF (over 300 km from the first case). The wild boar hunting activities in the bordering region led to the killing of two affected animals. This finding contributed to the implementation of disease control measures in this region, including the slaughter of over 100 free-range pigs and the placing of three neighboring villages in quarantine [7,18]. The last report of the disease came from a backyard farm on 26 October 2016 (36 domestic pigs slaughtered on 21 October 2016) in Valkivskiy region, although it is suspected that not all detected ASF cases in Ukraine are reported to OIE [33].

#### 2.2.4. Belarus

ASFV was first detected in Belarus in domestic pigs on a backyard farm in Grodno region on 21 June 2013 [3,7,17,25]. The second case of ASF was reported on 4 July 2013 in the Vitebsk region, over 400 km from the first outbreak. Since both outbreaks were reasonably distanced, the ASFV origin in both cases was unclear [33]. However, additional circumstances that might contribute to increased AFSV circulation and the high likelihood of ASF outbreaks in the EU member states are the well-connected wild boar population between Belarus and Poland, and Lithuania and Ukraine [11]. Moreover, even the low density of wild boar population in Belarus, which varies from three (Grodno region) to six (Brest region) animals per km^2^ [18], could result in the fast spread of ASFV to EU member states, where the number of wild boars is even higher. Although confirmed ASF cases in Belarus have been rare since in 2012, EFSA has submitted multiple media reports on ASF quarantine and depopulation procedures, both in domestic pigs and wild boars [7,18].

#### 2.2.5. Lithuania

Despite testing 10,430 domestic pigs and wild boars in 2013, Lithuania’s National Food and Veterinary Risk Assessment Institute (NFVARI) did not confirm the presence of ASF [35]. However, Lithuania’s State Food and Veterinary Service conducted a laboratory analysis on 24 January 2014, which confirmed the first ASF outbreak in two wild boar carcasses in Varena district. The first carcass was found 40 km, and the second carcass only 5 km from the border with Belarus [18,33]. Both cases were confirmed by the European Union Reference Laboratory for African Swine Fever (EURL) and indicated 100% sequence homology with the Belarusian ASFV type found in Grodno in July 2014 [12,18]. This suggests that the virus circulation correlates with the wild boar migration. The second ASF case was discovered on 24 July 2014 in a large domestic pig farm (with 19,137 animals), although routine serological testing for ASF one month prior to the outbreak had proved negative for ASF antibodies [18]. The farm had high biosecurity measures and was located in the Ignalina and Utena districts, about 170 km from the first Lithuanian ASFV cases in wild boar, suggesting that the outbreak was probably due to human involvement [18,33].

The epidemiological situation of ASFV should be regarded as a serious problem for Lithuania and other EU member states. A few reasons might have contributed to the increased frequency of ASF alerts in this region, including: the distribution of pig farms (Figure 1), the intensive swine production, and the density of wild boars—potential ASF vectors in this area. In 2010, the domestic pig population in Lithuania was estimated to be over 500,000 domestic pigs, located mainly in large breeding centers (around 60%), which include sows and around 400 other pigs [35], with established high on-farm biosecurity measures (Table 1, Figure 1). The small farm sector with over 25% of the domestic pig population and around 9% of pigs are located in backyard breeding systems with a high risk of ASF [36]. The latest reports have confirmed that the density of domestic pig population in Lithuania (Figure 2) is about 10–20 animals per km^2^, while the wild boar population in this country (Figure 3) is medium (0.6–1.1 head/km^2^) in Eastern Europe [18,37]. In summary, since the first outbreak of the disease in Lithuania in January 2014, there have been a total of 405 confirmed ASF cases in both domestic pigs and wild boars, included in 108 official OIE follow-up reports [32].

#### 2.2.6. Latvia

In Latvia, the first case was detected on 25 June 2014, followed by a rapid spread of ASFV. The disease was confirmed in three dead wild boars in the Kepova, Latgale region bordering on Belarus [32]. The next day, Latvia’s National Reference Laboratory confirmed three ASF cases in domestic pigs originating from a backyard farm in Robeznieku, Kraslavas region, located 6 km from Belarus and over 70 km from the Lithuanian border. The next few ASF outbreaks were confirmed in dead wild boars (16 July 2014), on an infected pig farm (17 July 2014; 58 animals), and another four ASF cases, considering contaminated grass as a potential source of the ASFV [18]. In July, the disease was also reported in the central part of Latvia (Madona region) and, in September 2014, new ASF outbreaks—in both domestic pigs and wild boar—were found in Kraslava (6 km from the Belarus border), Valka (close to the Estonian border), Ludza, and Rezeknes regions, 20 and 45 km from RF border, respectively [33].

Over 230,000 pigs in Latvia [36] are located in large farms (61%), small fatteners (13%), and 18% in other small farms (Table 1, Figure 1). In 2014, 32 ASF outbreaks were reported in the domestic pig population in Latvia, affecting 564 animals. Ten ASF outbreaks occurred the following year, affecting 213 domestic pigs. Fortunately, pig farmers reported nearly all suspected ASF outbreaks after the first clinical symptoms occurred, thus preventing outbreaks on a larger scale; the largest infected farm had 196 pigs. Other affected farms were mostly small backyard farms [18] with a low impact on pig production in this country.

In Latvia, ASFV spread fast in comparison to other nearby countries, affecting both the domestic and wild boar populations. Several reasons could have contributed to the fast spread of the virus, including Latvia’s geographic location; that is, being almost in the center of an infected area and bordering on affected countries. Other risk factors contributing to the spread of the disease were feeding pigs with kitchen waste and the large amounts of illegally disposed wastes in forested areas near human settlements [18]. An additional risk factor is Latvia’s high density wild boar population, reaching 1 animal per km^2^ (Figure 3). However, to reduce the ASFV circulation among the wild boar population in November 2015, the authorities implemented disease control measures by targeted hunting of the female population [37]. OIE has published 81 reports with a total of 1179 confirmed outbreaks in domestic pigs and wild boars in Latvia since the first ASFV case was noted [33].

#### 2.2.7. Estonia

In Estonia, ASFV was detected at several dispersed locations. The first case was on 2 September 2014, when a dead wild boar (piglet) was found in the Hummuli, Valga region (ASF buffer zone) 6 km from the Latvian border, and this was confirmed three days later. The location of the second suspected ASFV case in wild boar was reported on 7 September 2014 in the Tarvastu, Viljandi region (over 30 km from the first outbreak), and this was confirmed three days later. The third case of the disease in wild boar was found in the Lüganuse, Ida-Virumaa region on 14 September 2014 (confirmed 18 September), over 180 km and 150 km, respectively, from the first outbreaks and only 50 km from the RF highly forested belt [18].

The number of pigs in Estonia is smaller than in Latvia and Lithuania (Table 1, Figure 1) and only about 6% are located in small domestic pig farms, with low-level biosecurity (backyard and free-range breeding). Most of the swine production is concentrated with large breeders (72.7%) and large fattener (20.4%) farms [36]. Recent reports indicate that the average number of pigs is at least 10 animals per km^2^ (Figure 2) [36]. Fortunately, the density of wild boars in this area is lower (0.4–0.6 head/1 km^2^) than in Lithuania or Latvia, but higher than in RF (Figure 3) [18]. There have been, in total, 1052 confirmed ASF cases in both domestic pigs and wild boars, included in 92 official reports since the disease came into Estonia in September 2014 [33].

#### 2.2.8. Hungary

ASFV occurred in Hungary in April 2018 in Heves County (Figure 4) [39]. After that, in May, the ASF virus was detected in Szabolcs-Szatmár-Bereg County. The first wild boar was found only 1 km from Ukraine. The infected area has been arranged as a high risk area. In October 2018, in another Hungarian county, the ASF virus was noted. An infected area was created and up to 31 October there had been no outbreaks in domestic pigs and only 48 cases in wild boars. Probably, the origin of the infection was the natural expansion of wild boars from Ukraine, but there is still no confirmed evidence of the ASF source or the direction of the ASF spread in Hungary [39].

#### 2.2.9. The Czech Republic

In the Czech Republic in June 2017, two wild boars were found with the ASF virus [39]. This was in the Zlín District (Figure 4), where, after that, a quarantine area was imposed, including a ban on hunting. Only approved hunters who had received specific training on biosecurity could hunt. It was very important to prevent the spread of the virus. Up to 31 October 2018, there had been 230 cases in wild boars in the country. A year after the first ASF case, thanks to the actions of the authorities, the virus had been restricted to only a small area of the country. The feeding of wild boar was prohibited. The installation of dams and electric fences has taken place in the high-risk infected area. In addition, farm controls for biosafety have been applied [39].

#### 2.2.10. Romania

The ASF virus appeared in Romania in July 2017. Two outbreaks were proved in domestic pigs in Satu Mare County [39]. After that, in Tulcea County, the first case in wild boars was confirmed (Figure 4). These two areas had different spread dynamics for the virus. The changes in the second region were destructive. The virus could have been spread by hematopoietic insects (mosquitoes, flies) or weather conditions. By 31 October 2018, there had been 1073 outbreaks in domestic pigs and only 155 cases in wild boars. This situation was different from that in other counties; i.e., many outbreaks in domestic pigs and much fewer cases in wild boars. Most of these were observed close to the River Danube. ASFV has expanded into the northwest and southeast regions of Romania [39].

#### 2.2.11. Bulgaria

In July 2018, work began on the construction of a fence along the land border with Romania [39]. In August 2018, the first ASF outbreaks in domestic pigs were confirmed. This was in a small pig farm in Tutrakantsi, a village in Varna Region (Figure 4). The source of the infection was not known, but the village is located 100 km from Romania. In the Varna Region, the movement of pigs was restricted. Supervision and protection zones were set up. In addition, control measures were executed. In total, only one outbreak in domestic pigs and six cases in wild boars were confirmed in Bulgaria. Across the whole country, biosecurity and supervision increased and wild boar hunting took place, all year long. The import of wild boars into Bulgaria was prohibited [39].

#### 2.2.12. Moldova

An outbreak of ASF was reported on backyard pig farms in Moldova in July 2018. The outbreak affected two backyard farms in Donduseni [39]. Of the 13 pigs exposed to the virus, 11 cases were reported. The outbreaks are thought to have been caused by swill feeding (Figure 4). By December, all hotbeds of ASFV were completely liquidated in Moldova. There are still active hotbeds in Romania and Ukraine, so the level of vigilance will not be reduced. According to the legislation in force, the registered ASF virus in wild boars must be monitored during the next 24 months. These cases are localized in the districts of Rezina, Orhei, Cimislia, Cahul and Stefan Voda [39].

#### 2.2.13. Belgium

Initial cases of the ASF virus were reported in Belgium in September 2018 [39]. The remains of three boars were discovered close to the Bois de Buzenol, in the southeast of Belgium (Figure 4). The process of searching for the contaminated zone was begun by the Public Service of Wallonia and the Federal Agency for the Safety of the Food Chain, approved by the European Commission. By 31 October 2018, 132 cases had been noted in wild boars and none in domestic pigs [39].

## 3. ASFV Situation in Poland

ASFV notifications in Eastern European countries initiated implementation of numerous preventive campaigns in Poland. To prevent new disease outbreaks, a large-scale awareness campaign for citizens and travelers was started, together with training sessions for veterinarians, breeders and hunters. Simultaneously, the government implemented an ASF monitoring program that also marked buffer zones toward Belarus or Lithuania in the case of ASF virus spread. The program demonstrated that all tested animals (15,187, including 2124 domestic pigs and 13,063 wild boars) during the period 2011–2013 were ASF virus-free [42]. Moreover, the simulation exercise for ASF outbreaks (called “LIBERO 2013”) was practiced in 2013 to test the procedures for contagious diseases and the cooperation between the institutions responsible for ASFV control and eradication (Chief Veterinary Officer, Veterinary Inspectorate, Governmental Center for Security and Ministry of Agriculture and Rural Development) [42]. Although the simulated action and the cooperation between the national services were well prepared, ASF was detected in Poland in 2014.

The first ASFV outbreak was reported by the General Veterinary Inspectorate of Poland on 17 February 2014 [43]. The first genetic material with ASFV in Poland confirmed by the National Research Veterinary Institute in Pulawy (National Reference Laboratory) was found in a dead wild boar in a swampy area, about 100 m from a backyard farm in Grzybowszczyzna (a village in Szudziałowo Commune community, Sokolski County, Podlaskie Voivodeship), approximately 900 m from the Belarusian border [18,32,42]. The second ASF case was found in a fresh carcass of a wild boar near Kruszyniany (a village in Krynki Commune, Sokolski County, Podlaskie Voivodeship), 3 km from the border with Belarus and 15 km from the previous incident. Another ASF outbreak detected earlier was confirmed on 18 February 2014 [18,33,43].

After the first ASF case in Poland, governmental institutions initiated immediate preventive measures. Within five days, 623 pigs from 118 farms in 57 localities were examined and blood samples were tested, providing negative results. Simultaneously, samples from hunted wild boars within 40 km of the first outbreaks, near the Belarusian and Lithuanian borders, were also tested. Between 18 February and 16 April 2014, 3901 suspected ASF samples were examined (1033 samples from domestic pigs and 2868 samples from wild boars), all providing negative results [18,42]. Despite the preventive measures, a third ASF case in a wild boar was reported by the General Veterinary Office on 29 May 2014. A dead wild boar was found in Rudaki village in Krynki Commune, Sokolski County, near a previously infected area—Krynki Commune [18,43].

In 2014, an upward trend in the detection of ASF in wild boar population was observed. The intensification of virus detection was concurrent with increasing numbers of animals found dead in a single ASF outbreak. For example, groups of wild boar carcasses were found near Losiniany in Krynki Commune (three animals), Bobrowniki in Sokolka Commune (four animals), Luzany and Skroblaki in Grodek Commune (six animals each) and nearby Pilatowszczyzna village, where a group of 16 dead animals were found (8 October 2014), and this was the largest ASF outbreak in Poland. The majority of ASF cases in wild boars were found in Podlaskie Voivodeship, an area with a medium density of wild boars (0.3–0.6 head per km^2^; Figure 3). The infected voivodeship has been under constant monitoring to prevent the virus transmission to other Polish voivodeships with higher wild boar density, that is, Lubelskie and Warminsko-Mazurskie (0.6–1.1 head per km^2^; see also Figure 3).

Since the first ASFV outbreak in Poland, there have been 5333 confirmed cases of the disease in wild boars (number of dead wild boars was 465; number of hunted wild boars was 4868), with a total number of 31,488 infected pigs (Figure 4 and Figure 5.) [18,33,44]. The current situation regarding ASFV is presented in Figure 5.

Official reports do not indicate any direct contact between wild boars and domestic pigs, and no correlation has been found between wild boar density and infected areas [18]. However, the authorities in Poland have reported several ASF outbreaks in domestic pig holdings. For example, the first ASF outbreak in domestic pigs was detected in a small backyard farm (five animals) in Grodek Commune (Bialystok County, Podlaskie Voivodeship) on 19 July 2014, about 25 km from the first diagnosed case in a wild boar [33,43]. Similarly, the second and third outbreaks were noted on small farms with low biosecurity levels in Grodek Commune in Bialystok County (one pig) and Sokolka Commune in Sokolski County (seven pigs) on 8 August 2014 and 1 January 2015, respectively. However, the next two ASF cases were reported in large breeding holdings in Podlaskie Voivodeship. The fourth case was found in Hajnowka Commune (Hajnowski County) on 24 June 2015 and in Wysokie Mazowieckie Commune (Wysokomazowieckie County), where the number of pigs in infected farms were 273 and 566, respectively. There were, in total, 11 cases in the Podlaskie Voivodeship (cases no. 6–10, 12, 16–18, 22–23), infecting between 3 and 110 animals per case. Although buffer areas and control measures were established, the ASF outbreaks were reported in neighboring voivodeships—Lubelskie (case no. 11, 13–14, 19, 21) and Mazowieckie (case no. 15, 20) [18,33,45].

Since the first outbreak of ASFV in Poland on 17 February 2014, 23 cases involved domestic pigs, mostly in backyard and small farms, but these were stamped out relatively quickly and there was no evidence of virus persistence [18].

The majority of the 9 million pigs in Poland (Table 1) [36] are kept on small farms (66%) for traditional and cultural reasons. This is the highest number of pigs kept in small holdings, relative to the EU as a whole (Figure 1, Table 1). As previously reported, illegal swill and freshly harvested grass, especially in low biosecurity farms, are potential factors contributing to the spread of ASF in free regions, which leads to huge economic losses [7,17].

The density of domestic pigs (Figure 2) in the most infected area (Podlaskie Voivodeship) is lower than in other parts of the country (10–20 head/km^2^) [36], which provides a potential opportunity for successful ASFV eradication. The only impediments to the process are: (1) “human factor” (low biosecurity) and (2) free ranging wild boar populations with medium density in this Polish region (Figure 3) and surrounding bordering countries.

## 4. Challenges and Perspectives

Following the first ASFV circulation occurred in Europe in 1960s [3,17], it took over 30 years to successfully eradicate the virus from affected countries. The second spread of ASF in Georgia and in RF in 2007 [11,25,33] contributed to the virus migration and outbreaks in nearby regions, as well as the high probability of outbreaks in bordering countries. It has been reported that the efforts to contain ASF in Eastern Europe during the past years have not been successful. The disease has evolved to give rise to an even more intricate situation, reflecting a complex interaction between sanitary, economic, environmental and sociological factors in the region [7,10].

Small scale pig holdings and backyard farms have been considered to be high risk farms contributing to ASFV circulation and outbreaks [10]. These holdings have low or non-existent biosecurity measures, which increases the risk of virus introduction on the farm, mainly through secondary contaminated materials, such as human clothing, equipment, and fresh grass. Moreover, the use of food waste as a feed source for the pigs on these holdings increases the chances for ASFV emergence. During the first ASFV appearance in the affected countries, most of the domestic pig population was kept on large scale holdings, except for Malta, which had most of its domestic pigs (57%) in backyard holdings. However, in the second introduction of ASFV in the countries of Eastern Europe, excluding Estonia, a large proportion of domestic pigs are kept on backyard farms or small holdings. For example, of the 9 million domestic pigs kept in Poland, 66% (i.e., roughly 6 million) are kept on backyard farms and small holdings with minimal biosecurity measures. This latter point indicates a high-level risk for the domestic pig population in Eastern European countries and the possibility for the virus to circulate and persist for a long time.

The wild boar population has been an important factor in maintaining and circulating ASFV in the pig holdings and this is closely related to the outbreaks in domestic pigs [10]. The transmission mechanism of ASFV from wild boar to domestic pig populations has not been fully clarified, although several hypotheses propose direct contact, interbreeding and infected meat ingestion as possible routes of infection [40,46]. Regardless of the infection route, the wild boar population largely contributes to the spread of the virus in other countries, particularly in those with highly dense populations of wild boars. The epidemic in wild boar populations has mainly been detected in forested areas, where the eradication of the disease by reduction of these animals is practically impossible [7,33]. The control of ASFV through killing the wild life population has previously been adopted in Africa without any success and was characterized as “expensive, difficult and ineffective” [14]. Recently, the control measures for reducing the ASFV circulation in the wild pig population have been adopted in Latvia, mainly through targeted hunting of the female population [47]. Despite the multiple factors that contribute to maintaining and circulating ASFV, such as ticks, low biosecurity farms, and illegal trade of infected meat products, the wild boar population plays an important role in the virus dissemination [21]. Therefore, efforts should be directed toward developing and initiating more strategic and effective control measures that will reduce and eradicate ASF in the wild boar population.

The characteristics and the virulence of ASFV have not changed significantly since it was first detected in Kenia in 1909 [3,14]. However, ASFV sequencing has revealed that the virus circulating in Georgia belongs to genotype II and originates from Mozambique, Madagascar and Zambia [31], thus excluding wild boars as possible virus propagators. In addition, during attempts to produce a vaccine against the disease in Angola, low virulence ASFV had been released causing atypical clinical disease manifestation and its ability to persist longer in infected populations [14]. However, observations made by field clinicians and studies in laboratory conditions indicate that the circulating virus remains highly virulent [10]. For example, an outbreak in Poland reported 16 wild boars dead at one location. The multiple outbreaks in wild boar and domestic pig population indicate that both populations are at a high risk of exposure to the virus, which particularly concerns pigs kept on small scale holdings and backyard farms with low biosecurity measures. Therefore, training, education, and awareness of possible routes of infection, improving the on-farm biosecurity and the disease control measures through involving stakeholders such as regulators, producers and veterinary practitioners are of high importance. Moreover, the pig industry may be successful in controlling the disease and, as in the case of Brazil, could initiate positive transformation in the pig industry.

Although there is intensive ongoing research on ASFV vaccines, currently there is no effective treatment that can prevent the global spread of ASF [21,27]. Disease control is based on animal testing, especially in infected areas and when there is a suspicion of ASFV introduction to other regions or domestic pig farms. Over the years, findings from research on ASF have confirmed that the only method to eliminate the virus from the environment is eradication by the slaughter and safe utilization of all susceptible animals [7,19]. Therefore, most of the efforts in ASFV eradication programs in affected countries should be concentrated on limiting the pathways of virus transmission among both wild boar and domestic pig population. The most important ASF preventive procedures are: (1) quarantine and implementation of culling procedures in all affected regions [3,7,11,26], (2) reduction of affected wild boar populations, especially where a high density of these animals is well known [6,7,17], (3) containing economic and socio-cultural animal breeding with free ranging pig practices [3,7,14,19,21,32], (4) counteracting illegal movement of swine and pork products from infected areas [7,8,14,31], (5) prohibiting the use of kitchen waste and contaminated grass as feed [3,7,14,17], (6) intensifying control measures in small and large pig holdings by authorized veterinary personnel in infected areas and nearby regions [7,8,14,17], and (7) the introduction of informative campaigns for breeders and hunters [3,6,18].

## 5. Conclusions

This study presents a chronological overview of the occurrence of ASF in Europe, with almost five years of fully reported disease presence in Poland (2014–2018). Since the first outbreak was reported in Georgia in 2007, the epidemiological situation has changed rapidly in Eastern European countries and EU member states. Although multifaceted preventive and hygienic procedures have been implemented in different countries over the last six years, the ASF virus has spread over 1200 km affecting RF, Ukraine, Belarus, Lithuania, Latvia, Estonia and finally reaching Poland. Currently, the Polish buffer area (mainly Podlaskie Voivodeship) has stopped the spread of ASFV into other parts of the country, as no new outbreaks have been reported over the internal buffer borders since February 2014. Although ASFV persists in wild boar populations in Podlaskie Voivodeship, it is only a matter of time before the governmental institutions and hunting associations can achieve eradication of the disease. Similarly, another challenge for effective eradication programs are the Polish backyard farms in the eastern provinces with low biosecurity standards. Finally, there is a high possibility of ASFV transmission to other countries from various sources, but the present epidemiological situation in Poland should be considered to be an example for ASF disease containment and a priority animal health problem in Europe.

## Figures and Tables

**Figure 1 viruses-11-00310-f001:**
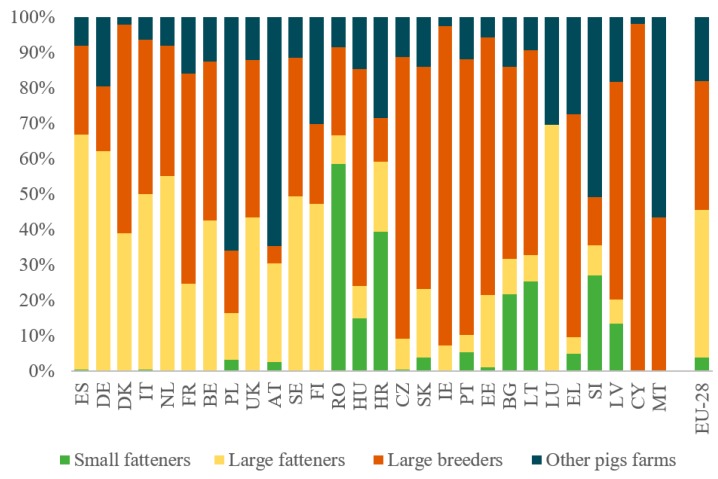
Distribution of pigs by the type of pig farm in the EU in 2014. Source [36].

**Figure 2 viruses-11-00310-f002:**
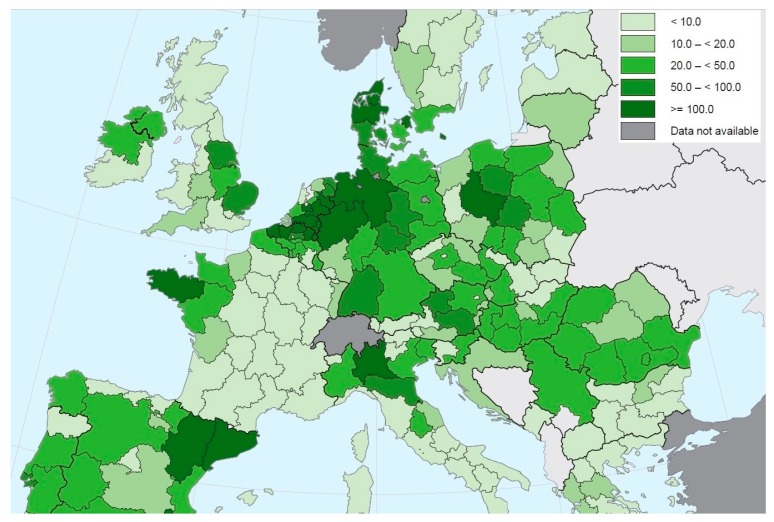
Domestic pig population (average number per km^2^) in Europe. Adapted from [37].

**Figure 3 viruses-11-00310-f003:**
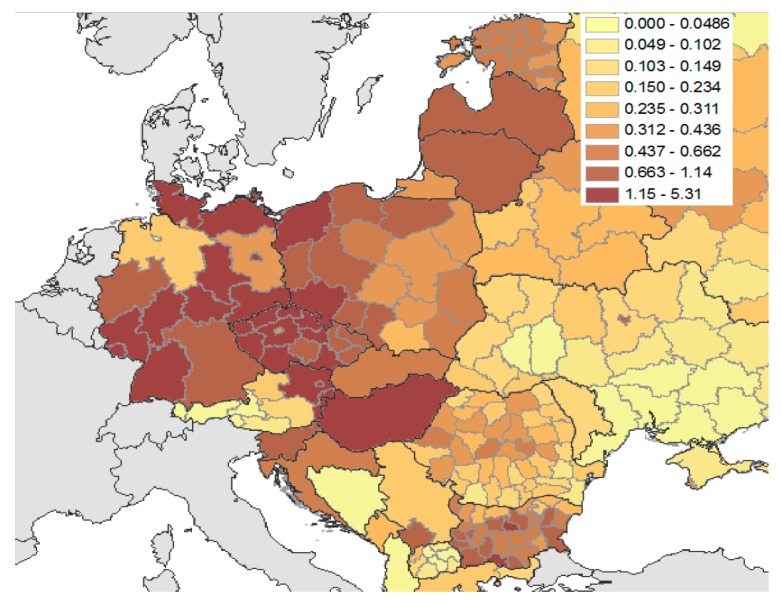
Wild boar population (average number per km^2^) in Europe. Adapted from [18].

**Figure 4 viruses-11-00310-f004:**
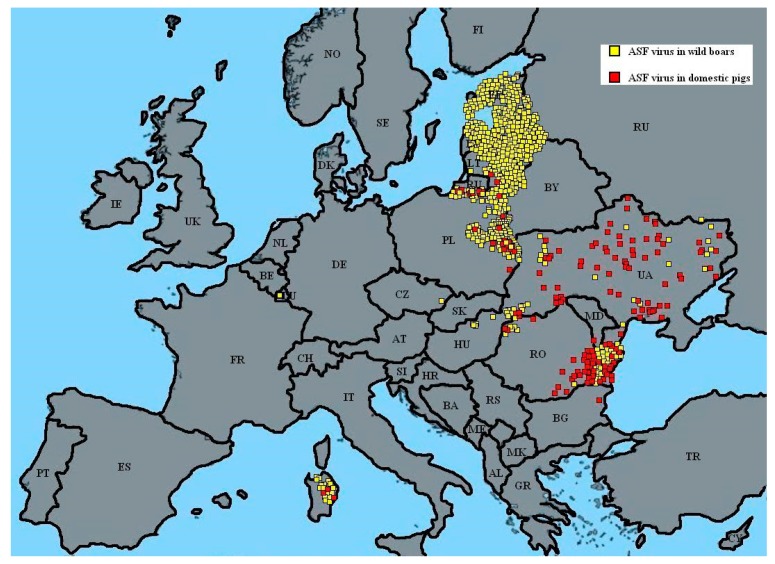
ASF situation in Europe (December 2018). Adapted from [40,41] (with the authors’ own elaboration).

**Figure 5 viruses-11-00310-f005:**
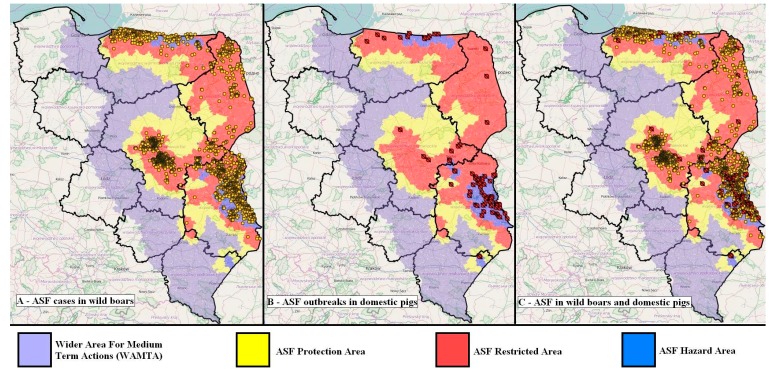
ASFV areas in Poland with a view of four infected voivodeships (December 2018). Adapted from [40] (with the authors’ own elaboration).

**Table 1 viruses-11-00310-t001:** Distribution of types of pig farms in the EU with the total number of pigs in 2018. Adapted from [36,38] (with the authors’ own elaboration).

EU Member States	Type of Pig Farm (%)				Total Number of Pig Farms by Countries	Total Number of Pigs (2018)
	Small Fatteners	Large Fatteners ^1^	Large Breeders ^2^	Other Farms ^3^		
Spain	0.5	66.4	25.1	8.0	15,534,530	30,000,000
Germany	0.3	61.9	18.2	19.6	16,582,160	27,600,000
Danmark	0.0	38.9	59.0	2.1	9,250,450	12,800,000
Italy	0.5	49.5	43.6	6.4	6,990,910	8,600,000
Neatherlands	0.0	55.1	36.9	8.1	6,037,490	12,300,000
France	0.2	24.5	59.2	16.1	8,486,880	1,100,000
Belgium	0.0	42.6	44.9	12.5	4,238,810	6,100,000
Poland	3.2	13.1	17.8	65.9	9,421,280	11,900,000
United Kingdom	0.2	43.2	44.5	12.1	2,771,700	4,700,000
Austria	2.6	27.8	4.8	64.7	2,079,370	2,800,000
Sweden	0.0	49.2	39.2	11.6	939,240	1,400,000
Finland	0.0	47.3	22.4	30.3	807,060	1,100,000
Romania	58.4	8.1	25.0	8.5	3,651,010	4,400,000
Hungary	14.9	9.0	61.4	14.6	2,108,190	2,900,000
Croatia	39.3	19.8	12.4	28.6	795,650	1,100,000
Chehia	0.4	8.8	79.5	11.3	1,156,840	1,500,000
Slovakia	3.8	19.3	62.8	14.1	359,880	600,000
Irleand	0.1	7.1	90.2	2.5	963,160	1,600,000
Portugal	5.3	4.8	77.8	12.0	1,078,280	2,200,000
Estonia	1.1	20.4	72.7	5.8	226,370	300,000
Bulgaria	21.7	10.0	54.3	14.0	456,040	600,000
Lithuania	25.3	7.4	57.8	9.4	512,070	600,000
Luksemburg	0.1	69.6	0.0	30.3	45,250	100,000
Greece	4.9	4.6	63.1	27.4	567,430	700,000
Slovenia	27.0	8.6	13.5	50.9	237,730	300,000
Latvia	13.3	6.9	61.5	18.3	232,470	300,000
Cyprus	0.3	0.0	97.8	1.9	188,200	400,000
Montenegro	0.0	0.0	43.3	56.7	45,700	no data
EU (28)	3.9	41.6	36.3	18.2	95,764,150	150,000,000

^1^ no. sows <10 other pigs (OP); ^2^ no. sows & at least 400 OP; ^3^ at least 400 pigs & 100 sows.

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
