# Peer review of "African Swine Fever Status in Europe"

_viruses, 2019, doi:10.3390/v11040310_

Round 1
Reviewer 1 Report
This review presents a chronological overview of African swine fever in Europe, from its 1st incursion and spread from the Iberian peninsular between 1960 and 1995 to the current events resulting from the introduction of the disease to Georgia in 2007, with a focus on the situation in Poland
The review describes details of usually the initial incidence of disease in a country and describes some of the relavent epidemiological situations relating to those countries. As such is it a useful collation of information on some of the ASF situations that have occurred within Europe.
A major problem with the manuscript however is that event with ASFV are moving rapidly and the manuscript is already out of date. There is no mention of events in Romania, Hungry, Bulgaria and Moldova, or the Czech republic or Belgium.
The text is also quite difficult to read and requires an extensive review of the English language and style, which in places conveys the incorrect information (ie was the Netherlands the "last" county (ie in terms of time ) to report ASF in the 1960-1995 period or was it the "remaining" country that was effected.
In the section on Spain and Portugal a single case in 1992 is described in some detail although it is unclear why this case is singled out.
The manuscript frequently details a total numbers of confirmed cases or other statistics for particular countries. However, as many of these situations are still ongoing it is unclear over what time period these figures relate.
The figures and tables are also lacking in information. for example what year do tables 1 and 2 refer to? Does the totals by country refer to farms or animals? The figures have no legend and figures 4 to 7 are not in English and are not adequately described in the text.
Author Response
Dear Reviewer,
At first, on behalf of the authors of the manuscript, I would like to thank to all of the comments and suggestions done by the anonymous Reviewer no. 1. We have done our best to correct the text as well as the figures according to given hints.
Answers to the revision:
1) The Reviewer comments related to the ASF disease and a fast spread of this epidemic status are right. It has to be pointed that every ASF epidemiological manuscript when published, will be already out of date. There is why we hope that our work will deserve to be accepted as a “snapshot” of the disease in Europe (described in two spreads: 1960 – 1995 and 2007 – 2018) and its outbreaks to the end of 2018.
2) The manuscript didn’t cover all the countries in eastern region, but our last focus was Poland – as a country were eradication of the ASFV is significant for the economy and livestock production in other EU Member States. Nevertheless, we strongly would like to meet the expectations of the Reviewer, there is why the manuscript was updated with the suggested informations about ASF situation in Czech Republic, Belgium, Bulgaria, Hungary, Moldova and Romania. Similarly all of the ASFV cases in wild boars and ASFV outbreaks in pigs in these countries were updated also on maps.
3) Section about Spain and Portugal was described in details as it was a first ASF situation in Europe. The reviewer's comments about too extensive description of this section were right, hence the description has been shortened what makes it more clear for the reader.
4) The chronological overview of previous ASF outbreaks in European Union was provided to show that previous outbreaks were a challenge to these countries but with applying specific measures the countries succeeded to control the disease. All of the dates in the text and in the descriptions of the figures and tables were updated. Moreover, it should be clarify that the Reviewer is right: ASF in Netherlands was not the “last” outbreak. This section about Netherlands was the last one (chronologically, in the first ASF spread), there is why we wrote about “last outbreak” what was an editorial clumsiness and what was changed in the manuscript.
5) Figures no. 4 – 7 were changed. Figure 4 presents all of the ASF cases in wild boars in European countries as well as the ASF outbreaks in domestic pigs in 2018. Similarly, detailed ASF situation in 2018 was presented in the Figure 5. The Reviewer also indicated doubts about the matrix of the map where some names were found in a language other than English, what is right. These maps are the only one project which is an official document of the Polish Veterinary Inspection and includes all of the ASF cases in Poland. There are no other official or governmental reports with a similar presentation of ASF outbreaks, so we kindly ask to accept our proposition of the maps.
6) The text was reviewed by an English native speaker, there is why all of the language mistakes are visible in the text ("Track Changes").
Please find the attachment (manuscript with corrections).
Kind regards,
Przemyslaw Cwynar

Reviewer 2 Report
Considering the active status of the ASFV disease, the authors should provide a more recent picture of the situation in Europe (e.g. Italy, Luxemburg, Romania, Bulgaria, Czech republic).
Author Response
Dear Reviewer,
Answers to the revision:
1) The manuscript didn’t cover all the countries in eastern region, but our last focus was Poland – as a country were eradication of the ASFV is significant for the economy and livestock production in other EU Member States. Nevertheless, we strongly would like to meet the expectations of the Reviewer, there is why the manuscript was updated with the suggested informations about ASF situation in Czech Republic, Belgium, Bulgaria, Hungary, Moldova and Romania. Similarly all of the ASFV cases in wild boars and ASFV outbreaks in pigs in these countries were updated also on maps.
2) The text was reviewed by an English native speaker, there is why all of the language mistakes are visible in the text ("Track Changes").
Please find the attachment (manuscript with corrections).
Kind regards,
Przemyslaw Cwynar

Round 2
Reviewer 1 Report
The modifications to include more recent information on additional countries and modifications to the figures have improved the manuscript, Unfortunately although the language has been reviewed it is still not close to an acceptable standard. In some places the revisions have not made scientific sense. The manuscript should be reviewed professionally for language corrections and also subsequently reviewed carefully by the authors to ensure that the scientific meaning is not altered.
Author Response
The authors of the manuscript would like to thank to the Reviewer no. 1. for the acceptance of methodological body of our work. Your valuable comments were significant and improved our study. Nevertheless, according to Your last comments we decided to correct the next using the assistance of professional service.
On behalf of the authors,
Przemyslaw Cwynar
Reviewer 2 Report
The manuscript at the present form, can now be accepted for publication in Viruses. The authors add substantial corrections.
Author Response
The authors of the manuscript would like to thank to the Reviewer no. 2. for the acceptance of our work. Your valuable comments were significant and improved our study. Nevertheless, according to the hints given by Reviewer no. 1. we decided to correct the next using the assistance of professional service.
On behalf of the authors,
Przemyslaw Cwynar